# Arginine-Enhanced *Termitomyces* Mycelia: Improvement in Growth and Lignocellulose Degradation Capabilities

**DOI:** 10.3390/foods14030361

**Published:** 2025-01-23

**Authors:** Wenhui Yi, Jingfei Zhou, Qiwei Xiao, Wujie Zhong, Xuefeng Xu

**Affiliations:** College of Food Science, South China Agricultural University, Guangzhou 510642, China; yiwenhui2729041972@stu.scau.edu.cn (W.Y.); zhoujfei@stu.scau.edu.cn (J.Z.); xiaoqiwei1117@stu.scau.edu.cn (Q.X.); zwj@scau.edu.cn (W.Z.)

**Keywords:** *Termitomyces*, arginine, mycelia, lignocellulose degradation, enzyme, transcriptome

## Abstract

*Termitomyces* mushrooms, known for their symbiotic relationship with termites and their high nutritional and medicinal value, are challenging to cultivate artificially due to their specific growth requirements. This study investigates the impact of arginine on the mycelial growth, development, and lignocellulolytic capabilities of *Termitomyces*. We found that arginine significantly promoted conidia formation, altered mycelial morphology, and enhanced biomass and polysaccharide content. The addition of arginine also upregulated the expression of the enzymes related to lignocellulose decomposition, leading to increased activities of cellulase, hemicellulase, and laccase, which accelerated the decomposition and utilization of corn straw. A transcriptome analysis revealed differential expression patterns of carbohydrate-active enzyme genes in arginine-supplemented *Termitomyces* mycelia, providing insights into the molecular mechanisms underlying these enhancements. The GO enrichment analysis and KEGG pathway analysis highlighted the role of arginine in transmembrane transport, fatty acid oxidation, and carbohydrate metabolism. This study offers a molecular basis for the observed phenotypic changes and valuable insights for developing optimal culture strategies for *Termitomyces*, potentially enhancing its artificial cultivation and application in the bioconversion of lignocellulosic waste.

## 1. Introduction

*Termitomyces* mushrooms, a group of saprophytic fungi known for their symbiotic association with termites, are primarily distributed in tropical and subtropical regions of Africa and Asia [1,2]. These fungi are highly valued for their exceptional flavor, high nutritional content, and potential medicinal benefits [3]. Despite their desirability, *Termitomyces* mushrooms are rare in natural habitats and have proven challenging to cultivate artificially [4,5]. Notably, they thrive on a specialized substrate within termite nests, the fungus comb, which is rich in lignocellulose—a complex matrix of cellulose, hemicellulose, and lignin [6]. Most of this specialized plant material consists of undigested plant residues from termites, which serve as the primary nutrient source required for mushroom growth [7]. The recalcitrance of lignocellulose to degradation is overcome by termites and various microorganisms present in the fungal comb, facilitating its absorption and utilization by mushrooms [8].

Edible fungi secrete an array of extracellular enzymes that degrade complex macromolecules during their growth phase. *Termitomyces* are known to utilize a diverse array of carbohydrate-active enzymes (CAZymes) and a select array of oxidative enzymes for biomass degradation [9]. Furthermore, transcriptomic analyses of symbiotic *Termitomyces* have revealed their genetic potential for lignin decomposition [10]. However, the capacity of *Termitomyces* mycelia to decompose wood debris in controlled environments appears to be limited, as indicated by the slow growth rate of the mycelia and challenges associated with accumulating substantial biomass [11]. Enhancing the lignocellulose decomposition capabilities of *Termitomyces* is essential for its artificial cultivation and potential application in the bioconversion of lignocellulosic waste.

Various nutritional and environmental parameters can be adjusted to enhance fungal growth and enzyme production to improve biodegradability. Previous research has examined the preferences of *Termitomyces* for different carbohydrates and varying carbon-to-nitrogen ratios during its growth. Amino acids, vital nutrients for fungal development, play significant roles in cellular metabolism, enzyme synthesis, and stress response; thus, they warrant further investigation in the context of mycelial growth and development in *Termitomyces*. Among these amino acids, arginine is recognized as a conditionally essential amino acid, noted for its diverse functions in fungal biology, including its role in the biosynthesis of proteins, nucleic acids, and polyamines, as well as the regulation of enzymatic activities [12]. However, the specific role of it in basidiomycetes, especially within the *Termitomyces* genus, remains relatively understudied.

In light of this context, we initiated a systematic investigation to evaluate the impact of arginine on the mycelial growth, development, and lignocellulolytic capabilities of *T Termitomyces* XY003. Through a screening process that encompassed 20 distinct amino acids, we identified arginine as a promising candidate for enhancing mycelial growth rate, biomass production, and spore germination. Subsequently, we further analyzed the effects of arginine on *Termitomyces*, focusing on enzymatic activity, biochemical parameters, differential gene expression, and mycelial catabolism for lignocellulolytic utilization. The results of this research enhance our comprehension of the role of nutrients in regulating mycelial growth and development in *Termitomyces*, offering valuable insights into strategies for improving lignocellulose decomposition and facilitating the artificial cultivation of this valuable fungus.

## 2. Materials and Methods

### 2.1. Strains and Culture

The strain of *Termitomyces* XY003 was isolated from Qianxinan, Guizhou Province, China, and stored in Applied Fungi Laboratory, College of Food Science, South China Agricultural University. *Termitomyces* was incubated in the following media at 28 °C and 120 rpm in triplicate. All samples were withdrawn periodically for triplicate measurements. Basal medium: 2% dextrose, 0.1% KH_2_PO_4_, and 0.5% MgSO_4_; control medium: basal medium with 0.1% (NH_4_)_2_SO_4_; amino acid medium (g/L): basal medium with 0.1% (NH_4_)_2_SO_4_ and 0.1% 20 different amino acids. Solid medium with 2% agar added to the above. All chemicals were of analytical grade. The starting pH was set at 6, with automatic adjustments made to maintain this level using 2 M hydrochloric acid and 2 M sodium hydroxide as needed.

### 2.2. Measurement of Physical and Chemical Indicators

Mycelial biomass was expressed as dry cell weight. At the end of fermentation, the mycelia were filtered, washed, and dried at 60 °C until a constant weight was obtained.

The colony diameters of the test strains on different amino acid media were determined using plate media, and the mycelial growth rate (mm/d) was measured and calculated using the crosshatch method (colony diameters were labeled every 2 d and incubated for 12 d).

A modification of the methodology proposed by Nguyen Van Long was implemented [13], using sterile water for the elution of *Termitomyces* spores. The resulting spore suspension was subsequently diluted to a concentration of 10^5^ spores/mL. A specific volume of this suspension was uniformly applied to a solid medium enriched with various amino acids. After 12 h, spore germination was microscopically assessed. The germination rate was assessed by calculating the proportion of germinated spores relative to the total number of spores observed in each microscopic field of view, which averaged 300 spores across five separate observations. Germination of a spore was confirmed when the length of the germ tube exceeded the diameter of the spore.

The filtrate was utilized for assessing the total reducing sugars via the dinitrosalicylic acid (DNS) assay [14], while the intracellular polysaccharide content was quantified using the phenol-sulfuric acid procedure [15]. Determination of L-arginine content occurred via Sakaguchi reagent method.

### 2.3. Measurement of Enzyme Activity

Cellulase and amylase activities were determined by the colorimetric method of 3,5-dinitro salicylic acid (DNS method) and expressed as the mass of reducing sugars formed per half hour catalyzing the hydrolysis of carboxymethylcellulose and starch at 50 °C and 30 °C, respectively [16,17]. Protease activity was determined by using casein as substrate. Lignin peroxidase (LiP) activity was assayed according to method described by Pinto’s method [18]. Manganese peroxidase activity was determined with reference to Li’s method [19]. Laccase activity was determined according to the method of Kumari and Das [20].

### 2.4. Structural and Chemical Analysis

*Termitomyces* was inoculated into corn straw containing arginine and without arginine addition and incubated at 28 °C. After 30 d, the corn straw was freeze-dried at −80 °C and then ground into powder.

The surface characteristics and microstructure of the corn straw were observed using a scanning electron microscope (FEI QUANTA FEG250, Europe, Eindhoven, The Netherlands). The samples were metal-sprayed, and SEM images were obtained at an accelerating voltage of 5 kV and 1000× magnification. Corn straw samples were finely ground in a uniform manner with KBr powder at a ratio of 2.5:100 to produce tablets. The absorbance of the samples embedded in the KBr matrix, as well as that of the external standard mixture, was recorded over the spectral range of 4000 to 400 cm^−1^ utilizing an infrared spectrometer (SHIMADZU IRTracer-100, Kyoto, Japan). The X-ray diffraction (XRD) method is based on the method of Chen et al [21]. Referring to NREL/TP-510-42623 and NREL/TP-510-42618, cellulose, hemicellulose, and lignin content in corn straw were determined by a two-step acid digestion method [22,23].

### 2.5. RNA Isolation and Sequencing

To further investigate the mechanism of mycelial promotion by arginine, we examined its transcriptional expression. Arginine-treated and no-arginine-treated mycelial sphere RNA was extracted using TRIzol lysate according to the manufacturer’s protocol. The RNA samples were then assessed for purity and quality using a NanoDrop spectrophotometer (Thermo Scientific, Nanodrop 2000, Waltham, MA, USA) and an Agilent 2100 LabChip GX (Agilent Technologies Inc., Santa Clara, CA, USA). Samples with RNA concentrations of at least 40 ng/μL, a volume of at least 10 μL, an OD260/280 ratio within the range of 1.7 to 2.5, an OD260/230 ratio between 0.5 and 2.5, and a normal 260 nm absorption peak were selected for sequencing library construction and subjected to 150 bp paired-end sequencing on the DNBSEQ-T7 platform. The initial sequencing reads were processed with SOAPnuke (v1.4.0) [24] to filter out reads containing adaptors, poly-N sequences, or low-quality segments, resulting in a set of clean reads. These clean reads were subsequently assembled using Trinity (v2.0.6) [25], and Tgicl (v2.0.6) [26] was applied for clustering the assembled transcripts and removing redundant information, thereby identifying a set of unique genes.

### 2.6. Gene Expression Analysis

Clean reads were aligned to the assembled unique genes utilizing Bowtie2 (v2.2.5) [27], and the gene expression levels were quantified through RSEM (v1.2.8) and subsequently normalized to FPKM (fragments per kilobase of transcript per million mapped reads). Functional annotation of the identified transcripts from XY003 was conducted by comparing them against the UniProtKB/Swiss-Prot protein sequence database, employing the best BLASTX hits. A default BLAST E-value cutoff of 10–5 was applied during the similarity searches. The Blast2GO suite facilitated the annotation of transcripts with Gene Ontology (GO) information [28]. The prediction of Carbohydrate Active Enzyme families was executed using the CAZymes Analysis Toolkit (CAT) [29], with further manual curation achieved by searching for homologies to previously annotated CAZymes within the NCBI protein database. Differentially expressed genes (DEGs) were identified using DEseq2 [30], with DEGs exhibiting a fold change greater than 2 or less than −2 and an adjusted *p* value of ≤0.001 deemed significantly differentially expressed. GO enrichment analysis and KEGG enrichment analysis were conducted using the Phyper, a function in R, with the significance levels of terms and pathways corrected by Q value, adhering to a stringent threshold (Q value < 0.05).

### 2.7. Validation of the RNA-Seq Results by RT-qPCR

To verify the accuracy of the RNA-seq data, RT-qPCR analysis was performed according to [31]. The primers used in this study were in Appendix A.

### 2.8. Statistics

The experiment was conducted in triplicate, and the results are expressed as mean ± standard error. Statistical analyses were performed using SPSS 16.0 software (IBM Corporation, Armonk, NY, USA) with a significance level of *p* < 0.05. All graphs were plotted using Origin Pro software (version 2021, Origin Lab Corporation, Northampton, MA, USA).

## 3. Results and Discussion

### 3.1. Screening of Vital Amino Acids for Mycelial Growth of Termitomyces

The findings derived from the experiments assessing the growth rate, biomass, and germination rate indicate that various amino acids facilitate the growth and development of *Termitomyces* mycelia to differing extents. Compared to the control group, mycelia exhibited the fastest growth in media containing arginine, glutamic acid, and aspartic acid. As illustrated in Appendix A the peripheral regions of the colony predominantly displayed a thin mycelial structure, while the central regions were characterized by the production of a substantial quantity of spores, which are crucial for the propagation of *Termitomyces* [32]. The presence of amino acids significantly enhanced the production of spores during the mycelia growth phase. In addition to observing spores on the plate, biomass serves as a more objective measure of the outcome of mycelial nutritional transformation. The data from Figure 1 showed that the mycelial biomass was lowest under nitrogen deficiency, a predictable outcome. However, when compared to the control group with ammonium sulfate, media enriched with arginine, glutamic acid, and aspartic acid yielded significantly higher mycelial biomass. While there was no strict one-to-one correspondence between biomass, mycelial growth rate, and spore germination rate, this discrepancy may be due to the inconsistent growth of vegetative mycelia and internal conidia. Overall, the biomass and mycelial growth rate trends were broadly consistent. In summary, arginine showed the most pronounced effect in promoting mycelial growth and may serve as an important nitrogen source for *Termitomyces* mycelia, leading us to conduct further experiments focusing on the role of arginine.

### 3.2. Effect of Arginine on Physicochemical Indices of Mycelial Growth

The influence of arginine on the mycelial morphology of XY003 was assessed through liquid fermentation and plate colony analysis. From Figure 2a,b, it is evident that the mycelia with the addition of arginine was in the form of larger spheres, and a large number of spores appeared on the plates. In contrast, the control group without arginine supplementation, as shown in Figure 2c,d, displayed a flocculent mycelial structure in the liquid medium and a relatively sparse aerial mycelial layer on the plates, with fewer mycelia balls formed. These observations suggest that arginine plays a crucial role in shaping the mycelial morphology and promoting spore formation in XY003.

As illustrated in Figure 3a, both treatments achieved biomass equilibrium approximately seven days post-inoculation, with the arginine-supplemented group exhibiting a biomass of around 4 g/L, which corresponds to a 44% increase compared to the control. Figure 3c further indicates that arginine enhances the synthesis of intracellular polysaccharides within the mycelia. The glucose content analysis in the culture medium, as shown in Figure 3b, revealed a more rapid decrease in the glucose levels in the arginine-treated fermentation broth compared to the control, suggesting heightened metabolic activity and improved glucose uptake and utilization by *Termitomyces*. Even after the biomass and intracellular polysaccharides reached an equilibrium after seven days, significant glucose consumption persisted, implying continuous metabolic processes within the mycelial cells. Additionally, the arginine concentration measurements in the liquid medium, depicted in Figure 3d, confirmed its consumption and utilization, highlighting the potential of arginine to serve as a nitrogen source or growth regulator that significantly impacts mycelial growth and development.

Arginine appears to play a pivotal role in the growth and development of Termitomyces mycelia. As a nitrogen source, arginine not only significantly enhances the biomass of the mycelia but also influences their morphological development, leading to the production of a large number of spores, which are essential for the reproduction and dissemination of the mycelia. Spores are a critical component in the symbiotic relationship between termites and *Termitomyces*, being indispensable for the spread of *Termitomyces* and playing a central role in the nutrient supply of termites and the stability of the symbiotic relationship. In addition to regulating the environmental conditions within their nests, termites cultivate *Termitomyces* through pre-digested substrates, and it is worth considering whether arginine is a key element in this process [33]. Furthermore, the rapid decrease in glucose levels in the fermentation broth treated with arginine suggests that arginine may enhance the metabolic activities of *Termitomyces*, increasing the efficiency of glucose uptake and utilization. The consumption and utilization of arginine further confirm its potential as a nitrogen source or growth regulator, which may involve complex molecular mechanisms, including changes in gene expression, regulation of enzyme activities, and accumulation of metabolic intermediates.

### 3.3. Characterization and Chemical Compositional Analysis of Corn Straw Structure

The degradation of *Termitomyces* mycelia resulted in alterations to the microstructure of corn straw. As illustrated in Figure 4, the surface structure of the untreated corn straw exhibited a compact and organized arrangement. However, following one month of degradation by *Termitomyces*, the previously compact structure of the corn straw was disrupted, leading to the emergence of numerous ruptured holes and fractured fibers. The impact of fragmentation caused by *Termitomyces* mycelia was particularly pronounced in the samples subjected to arginine treatment. These findings can be substantiated through the analysis of the physical and chemical properties of the straw.

Fourier Transform Infrared Spectroscopy (FTIR) is an analytical technique that leverages the vibrational frequencies of molecules to identify chemical bonds and functional groups. This technique capitalizes on the specific absorption of infrared light by these chemical bonds. The spectral region from 4000 to 2500 cm^−1^ is characterized by hydrogen bonding vibrations involving C-H, N-H, and O-H groups. The range of 2500 to 2000 cm^−1^ corresponds to the vibrations of triple bonds and cumulated double bonds. The 2000 to 1500 cm^−1^ region encompasses double bond vibrations, such as those found in C=O, C=C, and C=N. Lastly, the 1500 to 400 cm^−1^ range includes single bond vibrations, such as C-C, C-O, and C-N, along with the bending vibrations of the C-H and O-H groups [34]. FTIR quantification of chemical composition occurs by measuring the intensity and area of absorption peaks and analyzing changes in peak positions and widths. The infrared spectrum characteristic absorption peak and its attribution of corn straw are shown in Appendix A.

As shown in Figure 5a, the absorption peaks corresponding to the spectra of the corn straw were relatively consistent, although the intensities varied. The peak at 1613 cm^−1^ corresponds to the vibrational mode of the lignin aromatic skeleton, while the peak at 1641 cm^−1^ is associated with the vibrational mode of the C=O bond in lignin linked to the aromatic ring [35]. This observation suggests that the lignin present in corn straw has undergone partial degradation. Furthermore, numerous sharp peaks were observed within the range of 1057 to 1740 cm^−1^, with a notable increase in sharpness at 1740 cm^−1^ when compared to untreated corn straw. This phenomenon can be attributed to the stretching and vibrational modes of the C=O bond in lignin, which is connected to the phenolic hydroxyl group of the aromatic ring side chain. The action of lignin-degrading enzymes appears to disrupt the structural integrity of lignin biphenyls, leading to metabolism of lignin fragments. This process results in the cleavage of aryl ether bonds and Cα side chains, yielding intermediates such as ketone-type and phenol-hydroxyl-type organic acids. The sharp peak observed at 1057 cm^−1^ was indicative of the stretching vibration of the primary hydroxyl group within the cellulose structure [36]. This suggests that the degradation of corn straw is primarily driven by the cleavage of ether bonds associated with phenylpropane in the lignin structural unit as well as by the disruption of the side chains linking cellulose and lignin. The degradation process initially leads to the formation of carboxylic acids, which subsequently break down into amino acids and other compounds. This gradual depolymerization of lignocellulose results in the accumulation of monosaccharides, amino acids, and other small molecules, which have not yet been fully absorbed and utilized by the combretum, thereby contributing to the observed increase in peak intensity.

X-ray diffraction (XRD) characterization was employed to elucidate the alterations in the fiber structure of the cultivated material throughout the growth and developmental phases of the *Termitomyces* mycelia. Typically, the amorphous constituents present comprise amorphous cellulose, hemicellulose, and lignin [37]. Cellulose can manifest in a crystalline form known as crystalline cellulose, wherein a portion of the amorphous cellulose chain is organized into a crystalline structure. The crystallinity index (CrI), which is the ratio of crystalline to amorphous regions, is a key determinant of the digestibility of lignocellulosic biomass. As illustrated in Figure 5b, the crystal diffraction intensity at 2θ = 22.5° of the corn stover treated with *Termitomyces* mycelia showed a decreasing trend, indicating a decrease in its crystallinity. Specifically, the degree of crystallinity decreased from 27% to 18.9% in the *Termitomyces*-treated group as compared to the untreated control group. Further addition of arginine treatment reduced the crystallinity to 16.7%. This result suggests that *Termitomyces* was able to effectively degrade and utilize the cellulose within the crystalline region, and the addition of arginine appeared to accelerate this degradation process.

These observations not only confirm the activity of *Termitomyces* in degrading lignocellulosic biomass but also imply a potential role for arginine in enhancing the efficiency of cellulose biodegradation. The decrease in crystallinity may be related to the activity of cellulases secreted by *Termitomyces*, which are capable of disrupting the crystalline structure of cellulose and transforming it into a form that is more readily available to microorganisms.

The analysis of the chemical composition revealed that the initial contents of cellulose, hemicellulose, and lignin in corn straw were 38.7%, 30.0%, and 17.9%, respectively. Following a 30-day exposure to *Termitomyces* mycelia, these values decreased to 33.3%, 25.1%, and 15.3%, respectively. In the treatment group supplemented with arginine, the contents of cellulose, hemicellulose, and lignin were further reduced to 30%, 21.8%, and 12.8%, respectively. Consequently, the degradation rates of cellulose, hemicellulose, and lignin were enhanced by 8.52%, 11.00%, and 13.97%, respectively.

### 3.4. Enzyme Activity Assay

The enzymes produced by fungi play a crucial role in the degradation of lignocellulose, with catalase, laccase, and peroxidase (LiP and MnP) being particularly significant. To further investigate the secretion of lignocellulases, the enzymatic activity associated with lignocellulose decomposition by *Termitomyces* was assessed. Specifically, carboxymethyl cellulase is classified within the cellulase family; xylanase within the hemicellulase family; and laccase, lignin peroxidase, and manganese peroxidase within the lignin enzyme family. The results shown in Figure 6 indicated that treatment with arginine enhanced the activities of laccase, cellulase, and hemicellulase, suggesting that arginine facilitated the degradation of lignocellulose in corn straw by *Termitomyces*. However, the activities of lignin peroxidase and manganese peroxidase were minimal, showing no significant variation following treatment. This analysis, in conjunction with the previous findings on lignocellulose degradation, suggests that laccase may function synergistically with other enzymes, such as cellulase and hemicellulase, during the lignin degradation process, thereby enhancing the overall activity and efficiency of laccase. Additionally, a stimulatory effect on amylase activity was observed, while protease activity remained largely unaffected. These findings may expand the potential applications of arginine as a growth-promoting agent for *Termitomyces* mycelia.

### 3.5. Analysis of CAZyme DEGs Related to the Decomposition of Corn Straw in Arginine

The principal enzymes responsible for the breakdown of lignocellulose are designated as carbohydrate-active enzymes (CAZymes), which can be systematically divided into six specific categories: glycoside hydrolases (GHs), glycosyltransferases (GTs), polysaccharide lyases (PLs), carbohydrate esterases (CEs), auxiliary activities (AAs), and carbohydrate-binding modules (CBMs) [38]. To elucidate the gene expression patterns in *Termitomyces* under arginine-supplemented fermentation conditions, we compared the differentially expressed genes (DEGs) between transcriptome samples from the arginine-supplemented (Arg) and control (CK) groups, as shown in Appendix A and Figure 7. Among the four-hundred-eighty-seven CAZyme DEGs identified, the expression of one-hundred-thirty eight was significantly upregulated and could be classified into thirty-four families, including twenty GHs, three AAs, three GTs, three CEs, three CBMs, and two PLs. Notably, genes encoding glycoside hydrolases that break down cell walls were prominently expressed, particularly endoglucanases from GH families 6, 7, and 9, which are crucial for cellulose degradation. GH7 endoglucanases play a pivotal role in lignocellulose degradation, primarily by catalyzing the stoichiometric hydrolysis of cellulose in β-1,4-glycosidic bonds, generating new chain ends and thereby disrupting the crystalline structure of the cellulose substrate. The presence of a carbohydrate-binding domain (CBD) in GH7 endoglucanases enables them to bind tightly to the cellulose surface, facilitating the effective degradation of cellulose [39].

Cellulose, the most abundant component of plant biomass, is a highly stable polymer consisting of many glucose units linked by β-1,4 bonds [40]. Hydrolysis of cellulose requires the combined action of several types of enzymes, and the GH family contains enzymes associated with cellulose and hemicellulose degradation with a strong predominance, suggesting that *Termitomyces* has the ability to degrade other components of cellulose and hemicellulose and that the action of arginine presumably enhances this degradation [41]. There are also AAs (11 AA7, 7 AA1_2, and 6 AA1) that play a key role in lignin breakdown: 9 GT4, 1 GT28, 9 GT15; 7 CE9, 4 CE8, 3 CE5; 1 CBM22, 2 CBM19, 2 CBM1; and 11 PL8, 2 PL1_7, which are found in arginine transcripts. These results indicate that arginine promotes the expression of carbohydrates in *Termitomyces*.

### 3.6. GO Enrichment Analysis

Gene Ontology (GO) functional classification enrichment analysis was carried out on the differentially expressed genes (DEGs), specifically focusing on molecular function, cellular components, and biological processes (Figure 8). The DEGs were mainly enriched in nine GO terms in the category of biological processes (Appendix A), including cellular processes, metabolic processes, localization, and bioregulation. Significant enrichments were observed in transmembrane transport (GO:0055085), eicosanoid biosynthesis process (GO:0046456), and steroid biosynthesis process (GO:0006692). These findings suggest that arginine supplementation has a pronounced effect on transmembrane substance transport and fatty acid oxidation. In the cellular components category (Appendix A), DEGs were predominantly associated with four GO terms, specifically cytoplasmic ribosomes (GO:0022626), membranes (GO:0016020), and their downstream branches, such as membrane fractions (GO:0044425), membrane intrinsic components (GO:0031224), and membrane components (GO:0016021). It is hypothesized that arginine induces changes in the membrane composition of *Termitomyces* mycelia cells, enhancing membrane permeability and facilitating the secretion of extracellular enzymes. Regarding molecular function (Appendix A), the top two GO terms at level 2 were catalytic activity (GO:0003824) and binding activity (GO:0005488). These were primarily related to ion binding and oxidative pathways. The ion-binding-related entries included ferric ion binding (GO:0005506), cofactor binding (GO:0048037), tetrapyrrole binding (GO:0046906), and hemoglobin binding (GO:0020037). Those related to oxidative pathways included redox enzyme activity (GO:0016491), which is a subset of catalytic activity. The enhanced metabolism and accelerated glucose consumption in *Termitomyces* mycelia, as a result of arginine supplementation, further support these findings.

### 3.7. Analysis of Carbohydrate Metabolic Enrichment Pathways

The top 20 KEGG terms are mainly associated with biosynthesis and metabolism. Notably, as in Figure 9, the pathways involved in carbohydrate metabolism, including starch and sucrose metabolism (ko00500), glycolysis/gluconeogenesis, (ko00010), and pentose and glucuronide interconversions (ko00040), were significantly enriched at the arginine level. In particular, 81 genes were highly expressed in the starch and sucrose metabolism pathway (ko00500), such as g4002_i0, which encodes for cellulase, endoglucanase [EC:3.2.1.4], and beta-glucosidases [EC:3.2.1.21 and EC:3.2.1.22]. Additionally, g3059_i0 encodes for cellulose 1,4-beta-cellobiosidase [EC:3.2.1.91], which is associated with glycoside hydrolase (GH) families 7 and 6. The observed data correspond to the activity levels of CAZymes, implying their participation in the breakdown of lignocellulose during the developmental stages of *Termitomyces* mycelia.

Glycolysis/gluconeogenesis, a central metabolic pathway, provides energy and biochemical precursors that are essential for cellular functions [42]. Our KEGG analysis indicated that those genes encoding key glycolytic enzymes, such as pyruvate dehydrogenase [EC:1.2.4.1] (g2744_i4, g2744_i0, g6152_i1, and g6152_i0), pyruvate decarboxylase [EC:4.1.1.1] (g3288_i0), and aldehyde dehydrogenase (NAD+) [EC:1.2.1.3] (ALDH) (g4043_i0, g5312_i0, g5979_i0, g5144_i0, g4736_i0, and g4769_i0), as well as ethanol dehydrogenase [EC:1.1.1.1] (g5605_i0, g7245_i0, g5511_i0, g5934_i0, and g5346_i0), were significantly upregulated in the arginine-supplemented transcriptome sample. The pentose and glucuronide interconversion pathway (ko00040) includes genes encoding UTP-glucose-1-phosphate uridylyltransferase [EC:2.7.7.9], D-xylose reductase [EC:1.1.1.307, 1.1.1.430, and 1.1.1.431], L-iditol 2-dehydrogenase [EC:1.1.1.14], and D-arabinitol dehydrogenase (NADP+) [EC:1.1.1.287]. While these genes are not directly related to CAZymes, they are likely crucial for maintaining metabolic homeostasis and energy production in *Termitomyces*. A further correlation analysis revealed a significant positive correlation between the qPCR and RNA-seq results (R2 = 0.8134, *p* < 0.001; Appendix A).

## 4. Conclusions

In this study, the supplementation of arginine was found to promote the formation of spores in *Termitomyces* and to alter the morphology of the mycelia during liquid fermentation. Meanwhile, both the biomass and polysaccharide content were significantly enhanced. Furthermore, the addition of arginine was associated with an increase in the expression of those enzymes related to lignocellulose decomposition by *Termitomyces*. This led to a significant elevation in the activities of cellulase, hemicellulase, and laccase, which in turn accelerated the decomposition and utilization of corn straw. To elucidate the molecular mechanisms underlying these enhancements, we employed transcriptome analysis to assess the expression of the carbohydrate-active enzyme genes in arginine-supplemented *Termitomyces* mycelia. The transcriptomic analysis revealed the upregulation of 138 differentially expressed genes (DEGs) involved in lignocellulose degradation, primarily glycoside hydrolases from families 6, 7, and 9. The differential expression patterns observed provide valuable insights into the genes and pathways involved in the arginine response, thereby offering a molecular basis for the observed phenotypic changes. This information will also be instrumental in establishing optimal culture strategies for *Termitomyces*.

## Figures and Tables

**Figure 1 foods-14-00361-f001:**
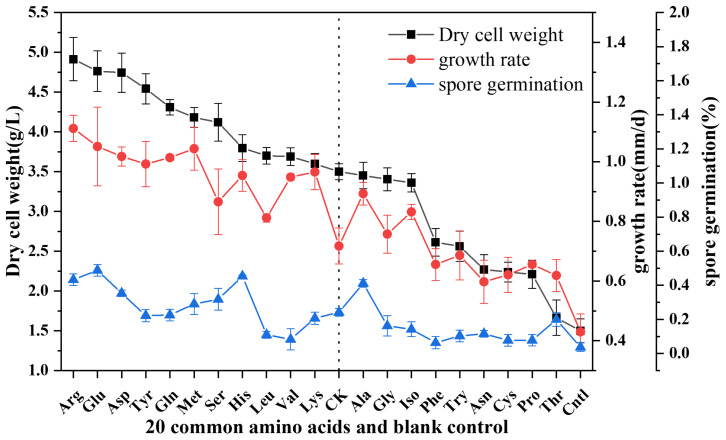
Effects of different amino acids on growth rate, biomass, and spore germination rate of *Termitomyces* CK is the control medium.

**Figure 2 foods-14-00361-f002:**
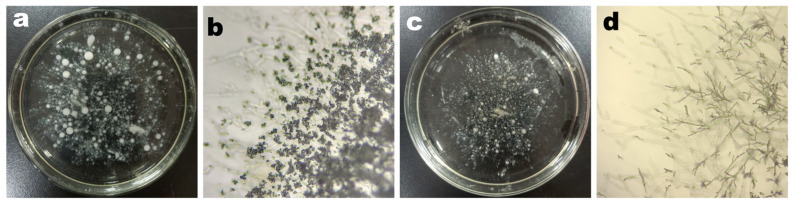
Macromorphology of liquid mycelia and micromorphology of plate mycelia. (**a**,**b**) The arginine-treated group; (**c**,**d**) the control group.

**Figure 3 foods-14-00361-f003:**
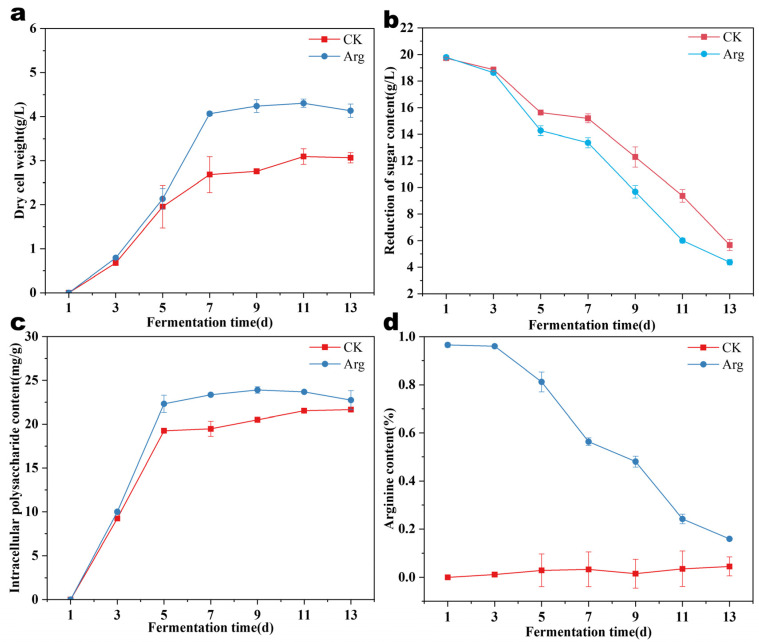
(**a**) Changes in biomass; (**b**) changes in reducing sugar content; (**c**) changes in polysaccharide content; (**d**) changes in arginine content.

**Figure 4 foods-14-00361-f004:**
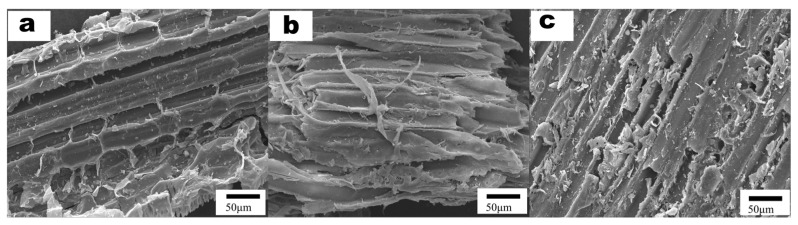
Scanning electron micrographs of pre-treated corn straw on the 30th day of mycelial growth ((**a**): untreated; (**b**): *Termitomyces* group; (**c**): *Termitomyces* + Arg group).

**Figure 5 foods-14-00361-f005:**
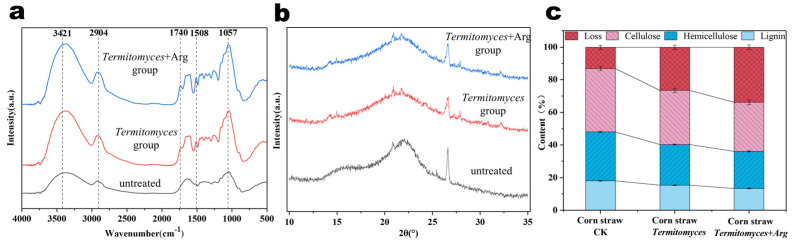
(**a**) Fourier infrared spectra; (**b**) X-ray diffractograms; (**c**) lignocellulose content of corn straw; data are mean earth standard error.

**Figure 6 foods-14-00361-f006:**
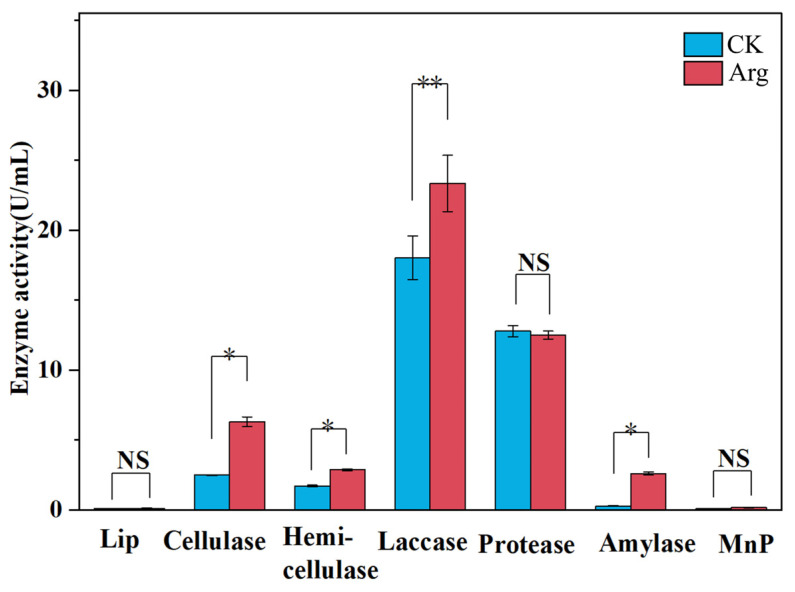
Effect of arginine on related enzyme activities. The asterisk ‘*’ denotes significant difference (*p* < 0.05), double asterisk ‘**’ denotes *p* ≤ 0.001, while ‘NS’ represents not significant.

**Figure 7 foods-14-00361-f007:**
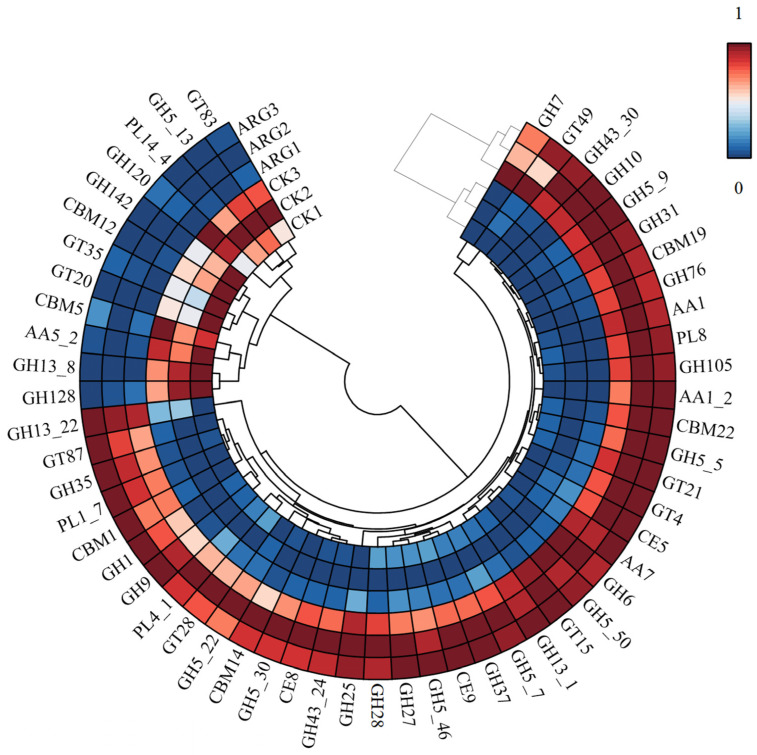
Heatmap of the effect of arginine on the predicted number of hits for representatives of different CAZyme families in the predicted proteomes of *Termitomyces*.

**Figure 8 foods-14-00361-f008:**
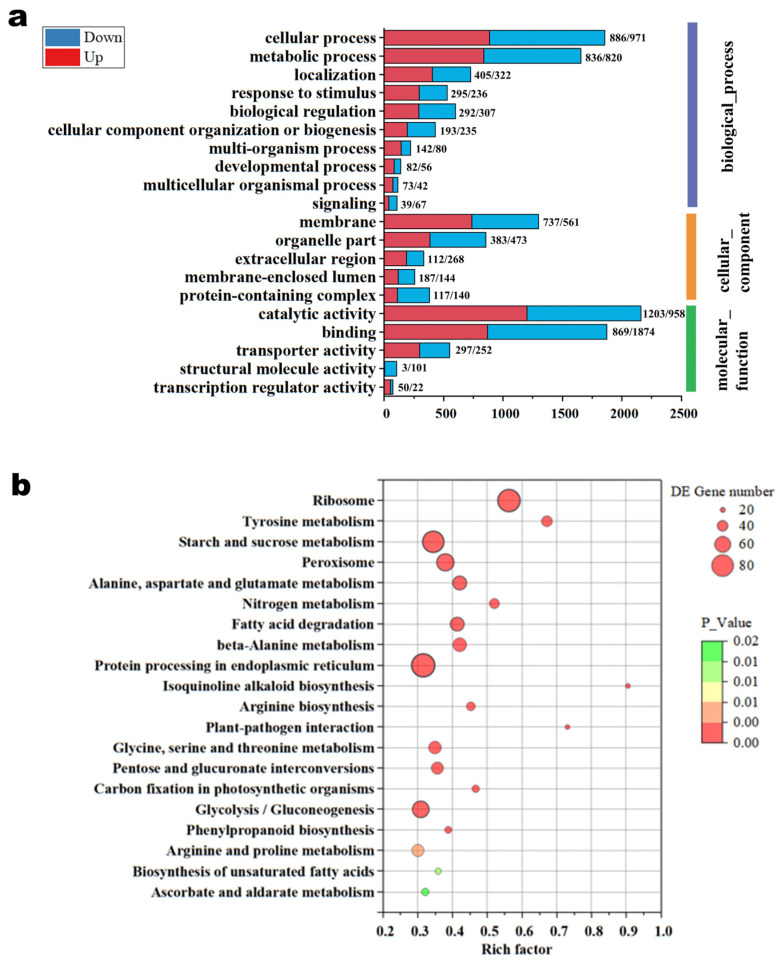
(**a**) GO classification of total DEGs in *Termitomyces* mycelia between the CK- and Arg-treated groups; (**b**) KEGG enrichment analysis in *Termitomyces* mycelia between the CK- and Arg-treated groups.

**Figure 9 foods-14-00361-f009:**
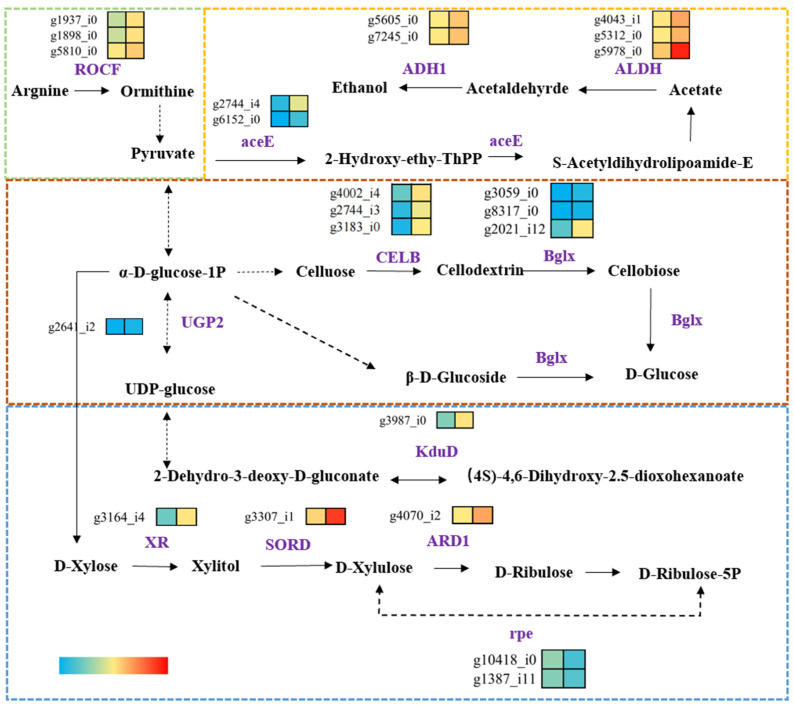
Reactions in carbohydrate metabolism, glycolysis/gluconeogenesis (ko00010), pentose and glucuronide interconversion (ko00040), and starch and sucrose metabolism (ko00500).

## Data Availability

The original contributions presented in the study are included in the article; further inquiries can be directed to the corresponding author.

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
