# Peer review of "Arginine-Enhanced Termitomyces Mycelia: Improvement in Growth and Lignocellulose Degradation Capabilities"

_foods, 2025, doi:10.3390/foods14030361_

Round 1
Reviewer 1 Report
Comments and Suggestions for Authors
The authors address a topic that is of interest to the journal's readers, and that also provides relevant information about ​​mushroom production. Authors are advised to review double spaces and null spaces between words. Greater clarity and quality are required in the figures, which are not adequately appreciated, mainly in figures 1, 3, and 5. Sections 3.1 and 3.2 do not discuss the results, and it is suggested that they be discussed. The references used could, in some cases, be more current.
Author Response
First and foremost, I would like to extend my sincere gratitude for taking the time to review our manuscript and providing valuable feedback. We deeply appreciate your professional critique and meticulous comments. Below, please find our detailed responses to your suggestions, along with the corresponding revisions, corrections, and tracked changes highlighted in the resubmitted document.
Comments 1: Authors are advised to review double spaces and null spaces between words.
Response 1:Thank you for pointing this out. I agree with this comment. Therefore, I've checked the full text and have removed the extra spaces, respectively, at Line228 423 530 in the latest update of the document.
Comments 2: Greater clarity and quality are required in the figures, which are not adequately appreciated, mainly in figures 1, 3, and 5.
Response 2: Agree. I have, accordingly, I have brought together the three tables in Figure 1 and sorted the data so that it looks much clearer. The other diagrams have been modified and replaced.
Comments 3: Sections 3.1 and 3.2 do not discuss the results, and it is suggested that they be discussed. The references used could, in some cases, be more current.
Response 3:Thank you for pointing this out. I agree with this comment. Therefore,I have added refinements to the discussion of results in Sections 3.1 and 3.2, marked in red at Line263-286. And some references of little relevance have been deleted, e.g. the original references 12,13,14, etc. Some references have been updated, such as reference 6. However, there are not many studies related to the ecological functions, physiological properties or applications of Termitomyces in these years, so there are not too many references to some of the older but landmark studies in the article, which have laid the foundation for the current body of knowledge. For example, reference 33, etc.
We are grateful once again for your valuable time and suggestions. We believe that with these revisions, our study is more rigorous and better positioned to contribute to the academic community. We look forward to your further feedback and are willing to make any additional modifications necessary.
Thank you again for your time and professional advice.

Reviewer 2 Report
Comments and Suggestions for Authors
The authors have studied mycelial growth, spore formation and germination, and corn straw degradation by Termitomyces with and without the s=addition of arginine.
The are several issues that need to be clarified in order to understand the relevance of the outcome of the experiments.
First of all, Termitomyces refers to a genus, question is what species the authors are dealing with. Is this known?
The authors are not very correct with references.
For example reference 12 is book on Chemistry and Amino Acids. I am not sure this is a proper reference to "vital nutrients for fungal development".
Reference 13 This reference deals only with the involvement of arginine during post harvest storage and not with all the functions mentioned here.
Reference 14 and 15 refers to the role of arginine in conidiation in actinomycetes but does not indicate that this amino acid plays a (similar) role in basidiomycetes.
Reference 35 is on Morchella and not on Termitomyces.
There might be more references that have to be checked.
In section 3.1 there is no reference to figure 1. In addition the legend of this figure is not readable and it is not clear for me what is the best amino aid and what is the control.
Furthermore, I like to know how the changes in Fourier spectra, X-ray diffractions, and enzyme activities were quantified. Arginine enhances the growth rate and increase the fungal biomass. So, without a good control, an increase in all the previous mentioned parameters are expected only due to the increase in fungal biomass. These might thus not be due to a simulation in degradation of lignocellulose by the amino acid but just due to a better growth of the organism. The authors should clarify this.
Although I cannot see in figure 1 what amino acids are best due to unreadable legends, I have the impression that more amino acids have a positive effect than only arginine. The manuscript is all about this amino acid but I question if this is or also true for other amino acids and maybe true for some type of amino acids.
Some minor remarks
Line 184: better “ fasted linear growth”
Line 193: what are bacterial filaments?
Author Response
Comments 1: Termitomyces refers to a genus, question is what species the authors are dealing with. Is this known? |
Response 1:Thank you for your inquiry regarding the specific species of Termitomyces used in our study. In our study, the experimental subject is an edible mushroom that lives in symbiosis with termites and belongs to the genus Termitomyces. This type of mushroom is primarily found in Yunnan, Guizhou, and Sichuan provinces in China. The strain used in this research was collected from Guizhou Province, China. Based on the sequencing results of the ITS region and morphological observations of the fruiting bodies, we preliminarily identified it as Termitomyces eurrhizus. However, as we are not specialists in species taxonomy, we cannot provide a definitive answer, which is why we have classified it to the genus Termitomyces. Fungi within this genus share commonalities in nutrient metabolism and ecological functions, so our research focuses on the genus Termitomyces. Our findings are significant for understanding the biological characteristics and potential applications of fungi in this genus.We appreciate your attention to this detail and hope this clarification addresses your concern.
|
Comments 2: The accuracy and relevance of some references were questioned. |
Response 2: Thank you for your insightful comments on the accuracy of literature citations in our manuscript. We have taken your feedback seriously and have recognized the concerns regarding the relevance of some references. During the writing process, we were indeed aware of the scarcity of literature specifically on the nutritional metabolism of Termitomyces, particularly studies focusing on arginine. As a result, we expanded our literature search to include more general fungal biology and nutritional metabolism research to provide a broader context for understanding the role of arginine in fungal development. Upon reflection, we agree that this approach may not have been entirely appropriate, and to avoid any ambiguity, we have removed the non-relevant references from our manuscript. We have taken steps to ensure that the remaining citations are directly related to our study and provide the most accurate and supportive background for our research findings.For example, reference 6 was replaced to better fit the article.
|
Comments 3:In section 3.1 there is no reference to figure 1. In addition the legend of this figure is not readable and it is not clear for me what is the best amino aid and what is the control. Response 3: |
Thank you for pointing out the issues related to Figure 1 in Section 3.1 of our manuscript. We appreciate your attention to detail and understand the importance of clear figure representation and legend interpretation. We appreciate your feedback and have taken immediate action to address the concerns raised. We have revised Figure 1 to enhance the clarity and readability of the data presentation. Specifically, we have: Improved the legend to ensure that it is now clearly readable and accurately reflects the treatments and conditions depicted in the figure. Reorganized the data within the figure to provide a more logical and straightforward visual representation of the results. Additionally, we have conducted targeted analyses of the results mentioned in Section 3.1, which has led to a deeper understanding of the findings. These analyses and their implications have been integrated into the main text, with the updated sections marked in red between lines 198 and 208 for your convenience. Regarding the controls, we have clarified under Figure 1 that the blank control (CK) refers to the group without arginine supplementation, while other amino acids serve as positive controls to assess their effects on fungal growth and sporulation.
|
Comments 4:Fourier Transform Infrared Spectroscopy (FTIR), X-ray Diffraction (XRD), and Enzyme Activity Changes Quantification Methods, as well as the Impact of Biomass on Experimental Issues. |
Response 4:Thank you for your insightful comments on the quantification methods for FTIR, XRD, and enzyme activity changes, as well as the impact of arginine on fungal growth and biomass in relation to lignocellulose degradation.In response to your concerns, we have implemented the following measures to address the potential confounding effect of increased fungal biomass on the parameters of interest: Quantification Methods: FTIR: We analyzed the intensity and area of absorption peaks, as well as shifts in peak positions and widths, to quantify changes in chemical bonds and functional groups. These principles are detailed in the manuscript, with specific references to the methodology highlighted in red between lines 336-348. XRD: The crystallinity index (CrI) was determined by measuring the intensity of crystal diffraction peaks, particularly at 2θ = 22.5°, and comparing these intensities to those of the untreated control group. Enzyme Activity: Enzyme activities were quantified using colorimetric methods, expressing the rate of substrate conversion to product under standardized conditions as international units (IU) per gram of biomass. We acknowledge the challenges associated with the increase in fungal biomass during the degradation of lignocellulose by Termitomyces, as well as the difficulty in controlling the intertwining of mycelium with corn straw. To minimize the interference of varying biomass on our results, we have taken the following measures in our sampling process: 1.Homogenization of Samples: We have ensured that the samples are thoroughly mixed to achieve a uniform distribution of mycelium and straw before taking aliquots for analysis. This approach helps to reduce the variability caused by the physical interaction between the fungal biomass and the lignocellulosic substrate. 2.Replication: To account for any inherent variability and to ensure the reliability of our measurements, we have performed the sampling in triplicate. This practice strengthens the robustness of our experimental data. Our experimental results suggest the potential involvement of arginine in the degradation and utilization of lignocellulose by Termitomyces. While the increase in biomass is a factor, it is not the sole driver of the observed effects. The subsequent transcriptomic experiments and enzyme activity assays provide additional evidence supporting the possibility of arginine's role in this process. We recognize that these findings are preliminary and open up new avenues for further exploration. We are committed to conducting more in-depth studies to elucidate the specific mechanisms by which arginine influences the lignocellulosic degradation capabilities of Termitomyces. This will involve a detailed investigation of the metabolic pathways and genetic factors that are modulated by arginine, and how these changes translate into the observed enhancements in enzyme production and substrate degradation.
|
Comments 5:The legend of Figure 1 was difficult to read, making it impossible to determine which amino acids had the best effect, and raised questions about whether amino acids other than arginine might also have a positive impact on Termitomyces. |
Response 5:We have updated Figure 1 to ensure that all amino acids are clearly labeled, and the legend is now easily readable. Our experiments have revealed that different amino acids exhibit varying degrees of superiority in promoting the growth of Termitomyces, and through comprehensive comparison, we have found that arginine is one of the more effective ones. This led us to focus our research on arginine as a potential breakthrough in identifying key nutritional factors that regulate the growth of Termitomyces. Our objective in this study is to use arginine as a starting point to discover critical nutritional factors that can control the growth of Termitomyces. Based on the findings of this experiment, we plan to conduct further studies on the combined regulation of key factors. We believe that understanding the synergistic effects of various nutrients, including amino acids, on the growth and lignocellulose degradation capabilities of Termitomyces will provide valuable insights for optimizing cultivation strategies and enhancing the biotechnological applications of this fungu. |

Reviewer 3 Report
Comments and Suggestions for Authors
In the manuscript entitled "Arginine-Enhanced Termitomyces Mycelia: Improvement on Growth and Lignocellulose Degradation Capabilities" the impact of arginine on the vegetative development and functional properties of Termitomyces fungi was investigated. The presented research results are a source of much valuable information, not only in relation to the importance of arginine in transmembrane transport and metabolic pathways of Termitomyces fungi, but also in relation to the development and subsequent optimization of artificial cultivation of these fungi. The cultivation outside the environment of natural symbiosis in this case is quite a challenge. Due to not only the nutritional value but also the medicinal properties, the search for factors that determine the effective growth of mycelium is extremely desirable. Hence, the presented topic, research problem and tips for its solution suggested in this manuscript will certainly be of interest not only to the scientific community but also to the industrial one. The right approach was to focus on the role of arginine, especially in the modulation of the growth and lignocellulolytic activity of Termitomyces mycelium. The results confirming its key importance in the proper functioning of the whole fungus organism.
The Authors have demonstrated great scientific maturity. The references (more than 40) are important and mostly up-to-date. Both the presented research methods and the way of conducting experiments, developing results and their presentation are appropriate. However, although the manuscript is of high quality from a substantive point of view, practically most of the graphics/infographics are illegible (also in the Supplementary file), so they must be improved before being accepted for publication in "Foods".
Author Response
Dear Reviewer,
Thank you for your insightful comments and for recognizing the value of our manuscript.You pointed out the lack of clarity in the use of some of the charts in our study. We have carefully considered your suggestion and have made changes to some of the images. Specifically, I have combined the three tables in Figure 1 and sorted the data so that it looks clearer. Figures 3 and 5 were also modified and replaced and Figure S2 improved the clarity.
We are grateful once again for your valuable time and suggestions.We look forward to your further feedback and are willing to make any additional modifications necessary.